# Fast Real-Time Data Process Analysis Based on NoSQL for IoT Pavement Quality Management Platform

Sung-Sam Hong [1], Jaekang Lee [2], Suwan Chung [3] and Byungkon Kim [3],*

1 Department of Multimedia Contents, Jangan University, Hwaseong 18331, Republic of Korea
2 Department of Civil Engineering, Dong-A University, Busan 49315, Republic of Korea
3 Department of Future and Smart Construction Research, Korea Institute of Civil Engineering and Building Technology, Goyang 10223, Republic of Korea
* Correspondence: bkkim@kict.re.kr

**Abstract:** The quality of road pavements is highly impacted by environmental variables, such as temperature, humidity, and weather; and construction-related variables, such as material quality and time. In this paper, an advanced data collection and analysis system based on big data/cloud was proposed for the use of IoT location-based smart platforms for pavement quality big data at road pavement sites. For the big data platform, a relational database management system (RDBMS) for a general alphanumeric data-based infrastructure for IoT-based systems was designed based on distributed/parallel processing to enable rapid and big data collection and analysis. The structure was established based on a NoSQL-based database to enable real-time high-speed collection and analysis, and the big data platform was developed as a data collection and visualization infrastructure. When the big data system was studied using data analysis methods, the proposed system demonstrated improvements in data collection performance and analysis speed, indicating that analysis results could be derived in real time. Specifically, the data collection processing (create) speed of the NoSQL-based system (0.405 ms) was significantly higher than that of the compared existing system (21.146 ms). Real-time processing capacity was also verified based on quality big data generated on actual road pavements, and the proposed system was proven suitable for the real-time monitoring (the data collection processing) of road pavement quality big data.

**Keywords:** road pavement; IoT; big data; database; data analysis

## 1. Introduction

Presently, the management of civil engineering, construction, and road construction sites is limited by the use of conventional analog-type processes. Although some construction projects that rely on conventional methods do progress according to established plans, problems such as frequent delays, false document processing, and poor construction continue to occur. In cases wherein the projects do apply Internet community technologies (ICT), the infrastructures are insufficient for monitoring and identifying these problems in real time. Otherwise, civil engineering and construction projects continue to be managed mainly through paper documents, handwriting, and communication via phone, radio, and messaging applications. Consequently, personnel and process management is inefficient, difficult to monitor, and dependent on the abilities of managers. Moreover, from a corporate or national viewpoint, management entities encounter difficulties due to lack of information or excessive paperwork for managing entire sites, identifying progress rates, and deploying suppliers. Civil engineering and construction sites have the environments with the lowest levels of digitalization and, consequently, lower productivities compared to those in other industries. Therefore, advanced ICT and digitalization are essential for the management of construction sites in the modern world.

The pavement management systems (PMS) of the future must be able to detect and predict all types of functional and structural defects, and be able to collect and analyze data

at various levels outside surfaces or inside packaging [1]. Smart construction technologies use data analysis techniques, which necessitate data collection and big data construction using Internet of Things (IoT) sensors [2,3]. These technologies are being researched for the automation (or semi-automation) of raw material management and process management in civil engineering and construction sites, which are currently performed manually. One important objective of these endeavors is to enable efficient process, quality, and risk management, while reducing unnecessary tasks and manpower. Combined with machine learning analysis, smart construction technologies enable various services such as image recognition, predictive analysis, determination of the quality of raw materials, and determination of appropriate construction timing and deployed equipment. For these purposes, pre-emptive data collection and big data/cloud construction using IoT are required. If the data generated throughout the construction cycle from raw material production are digitalized and the events that occur in construction sites can be analyzed based not only on structured data but also on unstructured data such as drawings, the management of construction quality and even transparency will be facilitated. Most IoT-based smart construction systems currently used in construction management are applied to processing, major work situations between processes, positioning of equipment or workers, and safety management based on the locations of equipment and workers. IoT location-based platforms have also been introduced to the construction industry for processes such as transportation and indoor navigation. Meanwhile, the construction of digital twin models for expansion to smart cities and their application to smart construction are being researched [4], whereas other research studies have proposed the use of dynamic quality monitoring systems (DQMS). For example, a DQMS for asphalt concrete pavements has been established based on the BeiDou Navigation Satellite System, intelligent sensing, IoT, and 5G technologies [5]. This system allows key technical indicators to be collected and transmitted throughout the entire asphalt mixture process flow, which includes manufacture in the mixing plant, transport in vehicles, paving, and compaction.

IoT location-based systems have advantages in locating and monitoring equipment or people in real time, analyzing data, and supporting decision-making. As the number of sensors is increased, larger amounts of data can be collected in short durations of time and stored in databases. However, this may cause difficulties in data retrieval and analysis. Because large data may lead to high computation and throughput, even simple statistical analysis may take a long time. In particular, for road pavement construction sites, increases in analysis time due to the diversity of mobile systems and multiple sites and devices may lower the applicability of IoT systems. Because pavement quality data is a typical big data due to the large amount of sensor data and very long construction period, new components of data management is required for a substitution for the relational database management system (RDBMS).

Therefore, this paper proposes an advanced data collection and analysis system based on big data/cloud to facilitate the use of IoT location-based smart platforms at road pavement construction sites. To enable advanced data collection and analysis, a big data platform was designed based on distributed and parallel processing for the RDBMS-based infrastructures used in conventional IoT-based systems. For this platform, a structure that can enable real-time high-speed data collection and analysis was established based on the non-structured query language (NoSQL). Subsequently, the big data system, which was designed to be able to perform real-time monitoring, was evaluated using data analysis methods suitable for road pavement construction environments. The proposed system provided improvements in data collection and analysis speed performance, thus solving the problem of insufficient data processing capacity typically encountered with conventional location-based systems and enabling analysis results to be derived in real time through advanced search and analysis. The performance of the proposed system was experimentally verified based on comparisons with a data collection and analysis system based on conventional RDBMS. The experimental results showed that the overall performance of the proposed system based on NoSQL was approximately 52 times (create) and 4.7 times

(delete) higher, and that its data collection (create) performance was particularly high. Therefore, it can be inferred that the proposed system, with its data monitoring and analysis system structure, is suitable for use in IoT road pavement quality management platforms that collect data on a large scale. In particular, the contributions of this research study include the following:

- Proposal of a smart system structure for road pavement quality improvement;
- Optimized data structure design for data pre-processing technology and road pavement process management that can determine road pavement status;
- Proposal of a framework capable of real-time process management through research and development of high-speed data processing techniques and protocols for IoT location-based road pavement quality big data;
- System structure capable of network traffic management as a component of smart city platform;
- Server scale-out technique that can be configured adaptively according to data scale.

The remainder of this paper is organized as follows: Section 2 provides details on related works in the literature. Section 3 describes the proposed IoT road pavement quality management system based on NoSQL. Section 4 reports on the experiments conducted to evaluate the performance of the proposed system. Finally, Section 5 presents the conclusions of the study.

## 2. Related Works

### 2.1. IoT-Based Construction Site Management Technologies

Thus far, research studies on the development of IoT-based construction site management technologies have focused mostly on safety and risk management, rather than on process management. Safety structure installation and worker control are often performed perfunctorily, and thus fall accidents tend to occur because of insufficient safety education and low levels of safety awareness in small construction sites [6]. Therefore, many studies have been conducted on the introduction of IoT-based smart systems for the prevention of such accidents. In 2019, Kim et al. [6] proposed a system that monitors workers who enter dangerous areas or streets and, when necessary, warns managers and workers of probable danger using cones installed on construction sites. This system prevents safety accidents by analyzing data collected through IoT cones using a long-term evolution (LTE) network. Figure 1 illustrates the structure of the system.

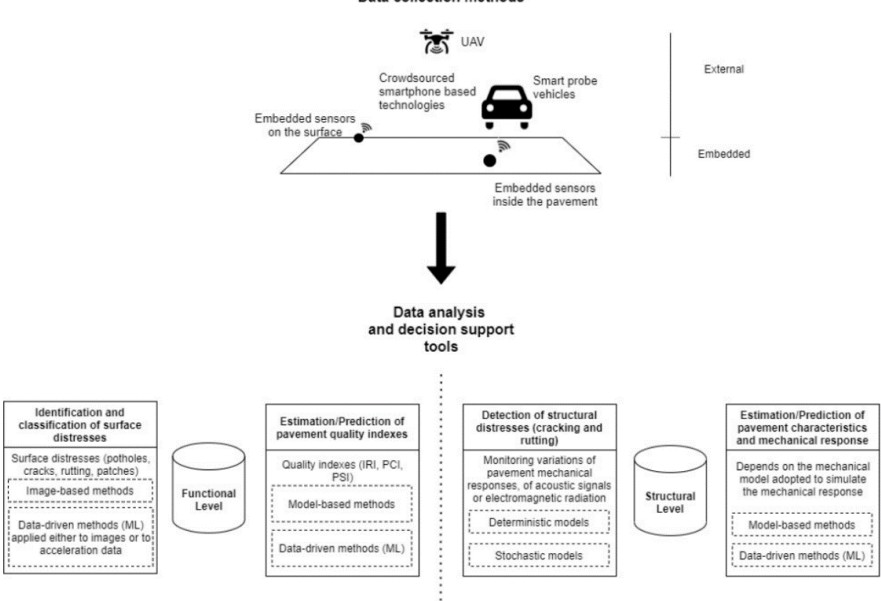

**Figure 1.** Decision support system for smart pavement [1].

In another study, an IoT-based cloud platform was used to manage prefabricated construction projects [7]. Therein, a model was proposed based on the interoperation of IoT and cloud for the collection and analysis of data-providing services throughout the entire process of prefabricated construction, ranging from production management services (production services) to transportation management services (logistics services) and construction site management services (on-site assembly services). This model, as a cloud asset data model, can efficiently show structured information in many areas, including production orders and logistics operations, by managing asset data collected from the construction process. Figure 2 illustrates the proposed model.

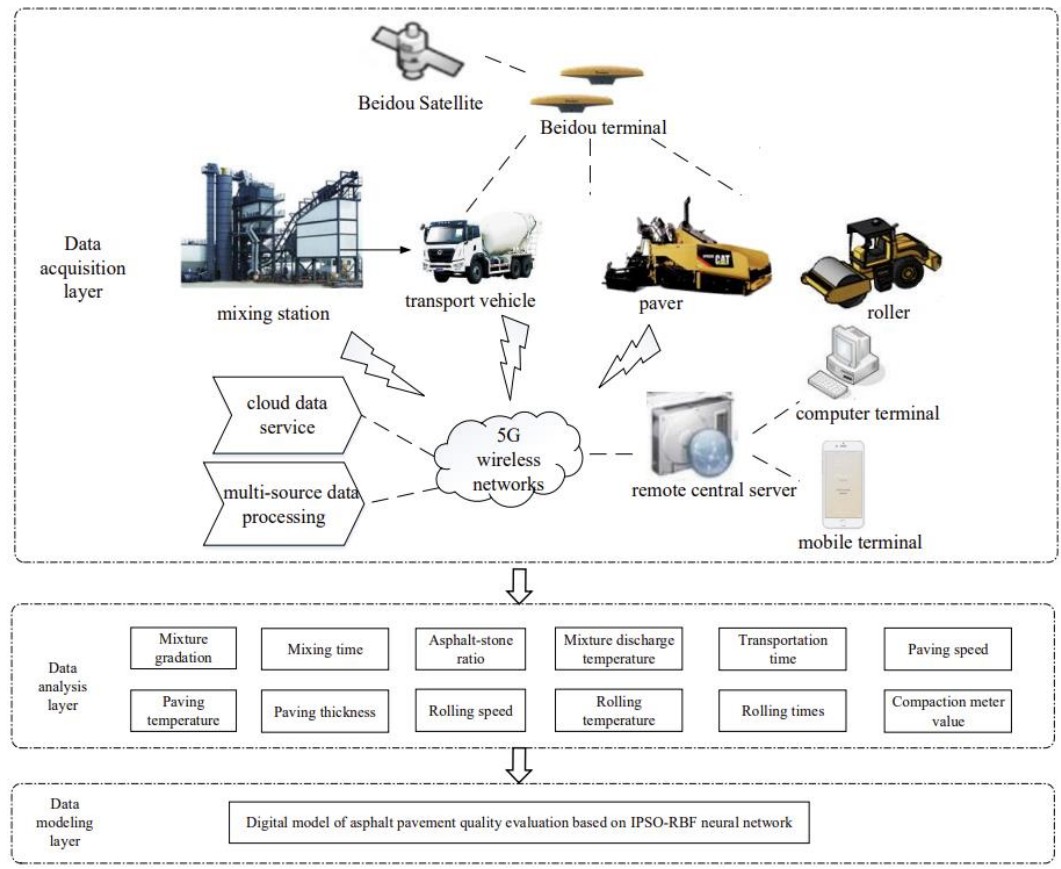

**Figure 2.** Structure diagram of digital monitoring system for measuring asphalt pavement construction quality [5].

Most of the other studies on construction site management using IoT were focused on safety management. In 2021, Kim et al. [8] proposed an Architecture for Building Smart Safety Management Systems Based on Construction Site IoT. A variety of predictive analysis algorithms were used to analyze the predictive types of safety prevention and build a smart safety management system. Later, a study on an IoT-based industrial site safety management system incorporating AI was also conducted [9]. As a research for quality control, there was a study on the development and demonstration of IoT-based DCPT (Dynamic Cone Penetration Test) technology for earthwork compaction measurement [10]. Recently, research using various IoT sensors is being conducted to develop an IoT-based system for concrete pavement quality control [11].

The sensors used include environmental sensors that can measure basic temperature, humidity, wind direction, and wind speed. Special sensors include an internal temperature sensor that can collect important information on asphalt quality, a compaction analysis sensor installed in the pavement, a curing measurement sensor for analyzing the quality of concrete roads, and an image collection sensor for visual analysis. [12] is to develop a road pavement aging model using a recurrent neural network (RNN) algorithm based

on the road pavement monitoring (RPM) data of the National Highway Pavement Management System (NHPMS) in Korea. In particular, environmental variables that were not considered in previous studies, such as average annual temperature and total annual precipitation, were considered when developing the model with RNN. Cloud-based smart building information modeling (BIM) technologies are also being actively researched [13]. Automated management can be obtained from building BIM data and project management based on a cloud. One study proposed a system that can perform smart management based on data mining by applying ICT to manufacturing process management, including raw material production, to address challenges such as various complex production processes, larger scale and uncertainty, more complex constraints, and a combination of different levels of operational performance [14]. Although studies on road pavement and construction process management are presently insufficient, construction ICT, particularly management technologies based on data analysis, is being extensively researched. However, research is still focused on RDBMS-based structured data collection or digitalization of offline documents [15].

### 2.2. IoT Sensing Device

IoT equipment and devices that can be used for construction site management vary according to their characteristics and purposes. For example, in the case of an image sensor, the image data acquired by the device can be used to monitor the status of the application of a concrete curing agent and to identify defective parts for construction quality management. For asphalt construction management, equipment based on a ground-penetrating radar (GPR) sensor can be utilized for the measurement of asphalt front compaction. For the construction management of cement concrete pavements, smart temperature and humidity sensors are used to manage the condition in real time. Additionally, information necessary for construction site management can be acquired using various IoT sensors such as global navigation satellite system (GNSS) sensors, integrated climate sensors, infrared cameras, acceleration sensors, and gyro sensors.

Through the use of these IoT sensors, it is possible to automatically manage resources and take immediate action in unexpected situations. Another advantage is that data acquisition and analysis can be performed quickly and accurately. However, there remain disadvantages in that some initial investment costs may be required and that problems with information security and compatibility will have to be addressed.

### 2.3. NoSQL

NoSQL (original meaning: non-SQL or non-relational) provides a mechanism for data storage and search using a less restrictive consistency model than that of the traditional relational database. The motives behind this approach include simplification of design, horizontal expandability, and detailed control [16]. NoSQL is a highly optimized key–value storage for simple search and additional tasks, and aims to achieve high performance gains in relation to latency and throughput. Thus, it is widely used in the commercial utilization of big data and real-time web applications. Furthermore, it can be used with an SQL-like query language, which can help make development more familiar; as such, it is also called "Not only SQL" to emphasize this feature [17].

However, as shown in Figure 3, NoSQL is different from SQL, which maps information based on the relations between two-dimensional tables. There are many different types of NoSQL because they encompass "not-relational" (complementary set) databases. Representative examples include key–value, which uses an associative array; column-based, which stores data in columns instead of rows as in conventional databases; and document-oriented, which uses JavaScript Object Notation (JSON) or Extensible Markup Language (XML) as the data format. The major characteristics of NoSQL as a database are as follows [18]:

- Suitable for semi-structured and unstructured data: NoSQL is useful for storing unstructured data. To express the hierarchical structure of the data, the database can be organized in a tree type or expressed as abstract graphs.
- Advantageous for storing large volumes of data: NoSQL can easily store large volumes of data because of its very nature, which is advantageous for distributed computing because only machines need to be added to the cluster.
- Excellent for solving problems for a specific domain: NoSQL stores data in data types of key values and graphs. As a result, it can achieve a high performance in a specific domain.
- Diversity of data-querying application programming interfaces (APIs): NoSQL can use various query languages by type and product. Also referred to as UnQL (unstructured query language), the query languages used with NoSQL are mostly lower than SQL, and thus complex queries can be difficult. When complex queries are required, the data structure may be massaged, multiple queries may be overlapped, or, in some cases, the ideas of SQL can be borrowed.

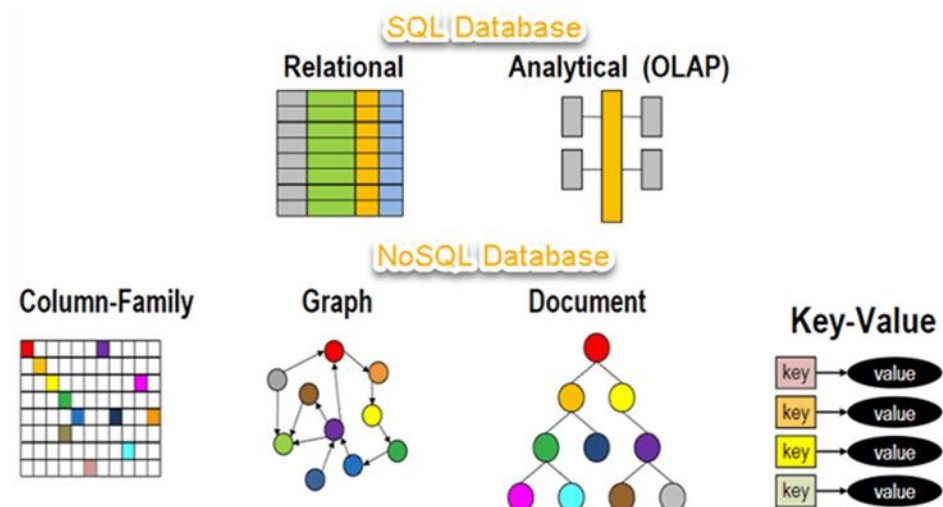

**Figure 3.** SQL vs. NoSQL [19].

NoSQL can search and store various types of information in the web environment but does not guarantee data integrity and accuracy. Furthermore, it does not use modification and deletion, which are replaced with input; does not require strong consistency; and is flexible for the addition and deletion of nodes and data distribution [20].

## 3. IoT Road Pavement Big Data Collection and Analysis System Based on NoSQL

### 3.1. Structure of Proposed System

The road pavement quality management and analysis proposed in this study consists of a road-pavement big-data system architecture for storing, processing, and analyzing location-based data and sensing data collected from IoT sensors. The overall architecture of the proposed system is illustrated in Figure 4. It is configured in accordance with basic big data layers consisting of data collection, processing, storage, analysis, and visualization structures. The collection model was designed and constructed using the representational state transfer (REST) API-based Python Web Framework [21] and web application [15].

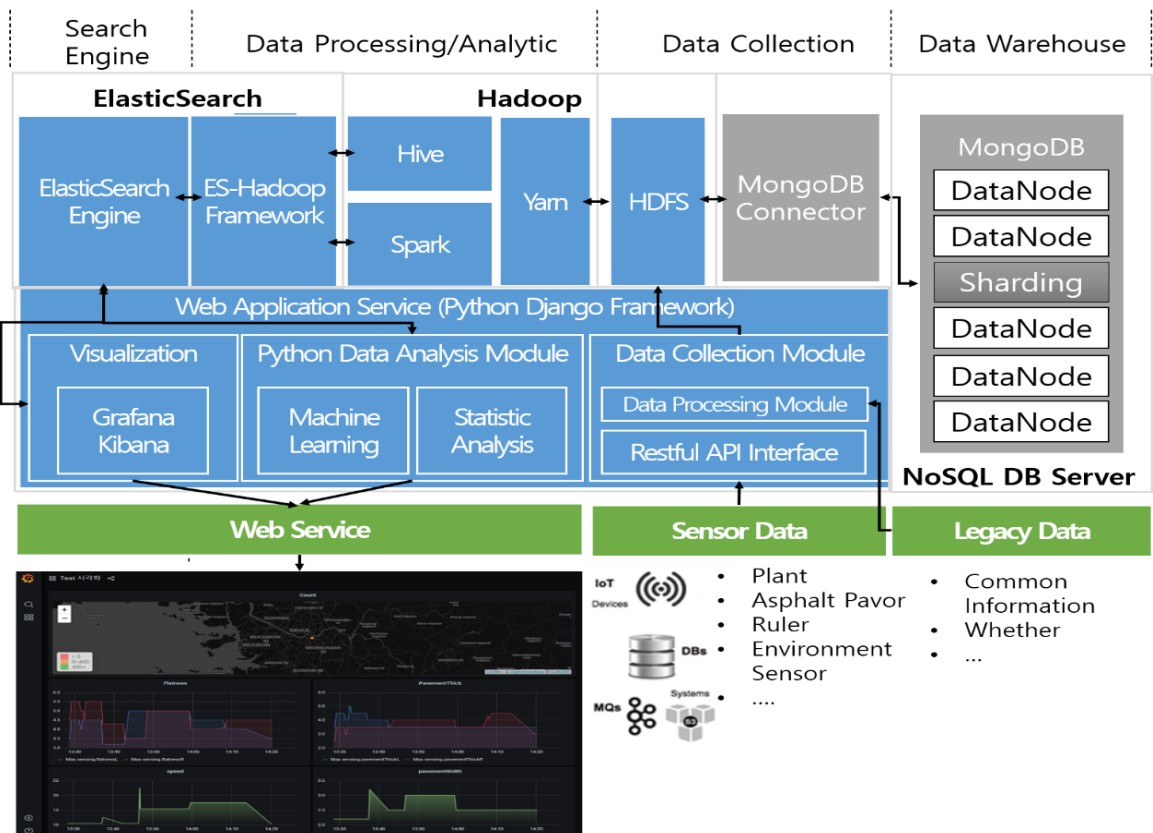

**Figure 4.** Proposed system.

When IoT devices send data via REST API, the collection model simply processes and stores them in the NoSQL database [22,23]. The protocol is simple, and storage processing is simplified by the use of the JSON format for the transmission data. The storage model stores data through distributed processing using the NoSQL driver interface. The data are stored in document format in distributed data nodes, and metadata for indexing are generated. The search and analysis module finds documents based on required keywords for post-indexing; this module can search at high speeds and performs statistical operations by applying distributed parallel processing in the form of MapReduce. When a web server and a web application are included in the model view template (MVT) structure, the web application also collects, processes, analyzes, and visualizes data [24].

The proposed system employs ElasticSearch [25] for data storage/management and high-speed search/analysis. Furthermore, the system architecture was designed such that high-speed data search and real-time data monitoring and analysis can be further facilitated via expansions in the data warehouse with MongoDB, a big-data-based NoSQL database structure. Because the proposed system can collect and process various heterogeneous data in real time from multiple road pavement sites, it uses a NoSQL-based data warehouse, which is not dependent on data format and is easily expandable. In addition, real-time analysis and monitoring were enabled with the use of high-speed search and data-processing technologies. Although this is also possible in conventional RDBMS-based data platforms, NoSQL, as a document-based database, is advantageous for data collection and processing in environments in which heterogeneous data and unstructured data sent from various heterogeneous devices coexist. MongoDB [26] is suitable for this purpose because it is easy to implement in the form of REST API to interoperate with an open-source-based big data platform such as Hadoop and ElasticeSearch, and also to interoperate with the open-source visualization platform Grafana. The early system ElasticSearch can be modularized using analysis/search engines, and a Hadoop–MongoDB system can be constructed in a form most suitable for road pavement quality management.

The system proposed in this paper is composed only of elements required in a commercial cloud environment in accordance with the definition and design of the functions and elements of cloud services. Hence, its structure is expandable to a stack environment, such as Elastic, Hadoop, or MongoDB, and is directly applicable to various platforms.

### 3.2. REST API Design

The sensors and data sources, REST API [27], and basic protocol for data transmission and reception between the web application server (WAS) and database (DB) server are shown in Figure 5. When the sensor data and collected data in JSON format in accordance with the data specification are sent via REST API, the data are parsed by the web application. These data are stored in the NoSQL DB using the Elastic environment in conjunction with the Elastic driver interface. The NoSQL DB, in which data are stored in the document data type, is suitable for this system because it can store data regardless of the format. The system then searches for and analyzes necessary information from the stored data.

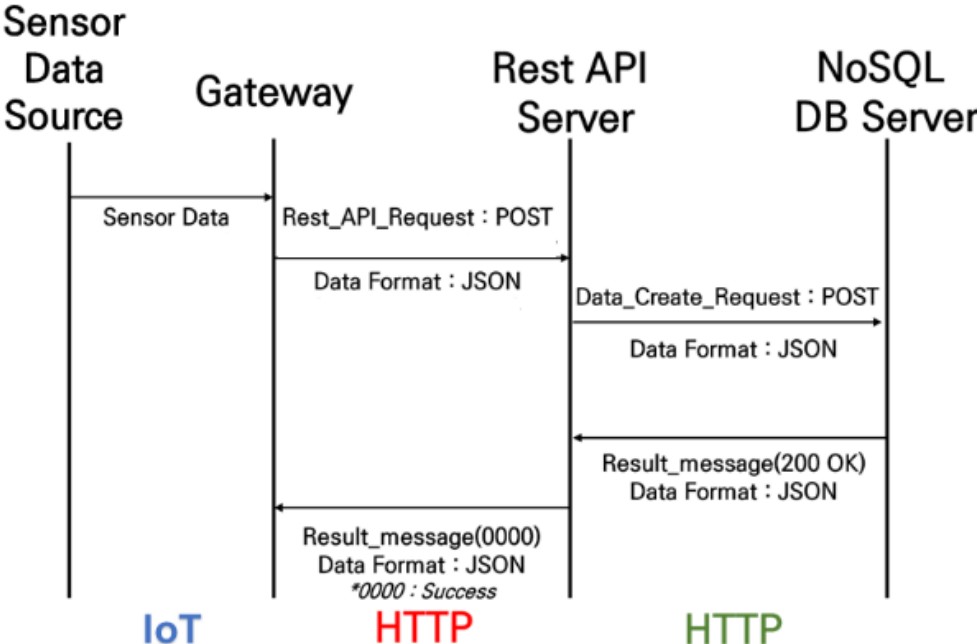

**Figure 5.** REST API protocol.

### 3.3. Development of Real-Time Road Pavement Quality Management Analysis System

The proposed system with the designed structure was developed to collect and analyze data in real environments, specifically, road pavement construction sites. The API was implemented to collect large volumes of data transmitted from IoT sensors in real time. In addition, the system was equipped with data visualization and analysis functionalities. Figure 6 shows the API implemented for use in a real road pavement environment.

It was demonstrated that the proposed system can collect data in real time from many sites and has a structure that is suitable for high-speed processing of big data in a road pavement quality management environment, in which large volumes of incoming IoT sensor data must be processed. Figure 7 shows real screens of the system dashboard for monitoring the collected data.

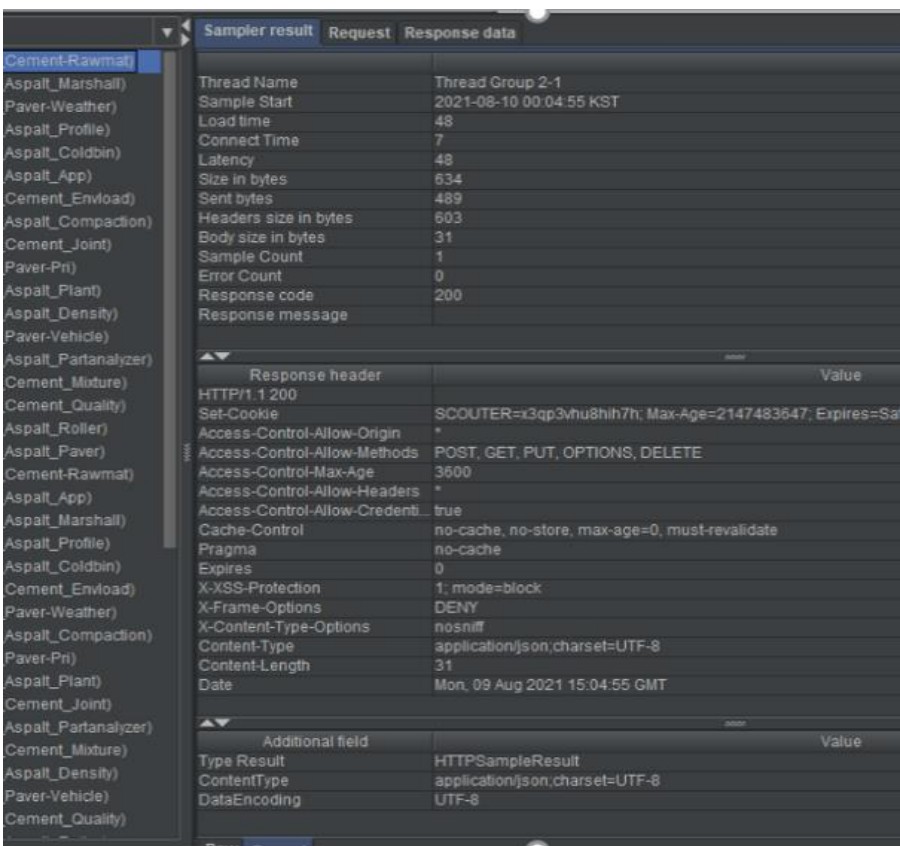

**Figure 6.** Interface API implementation.

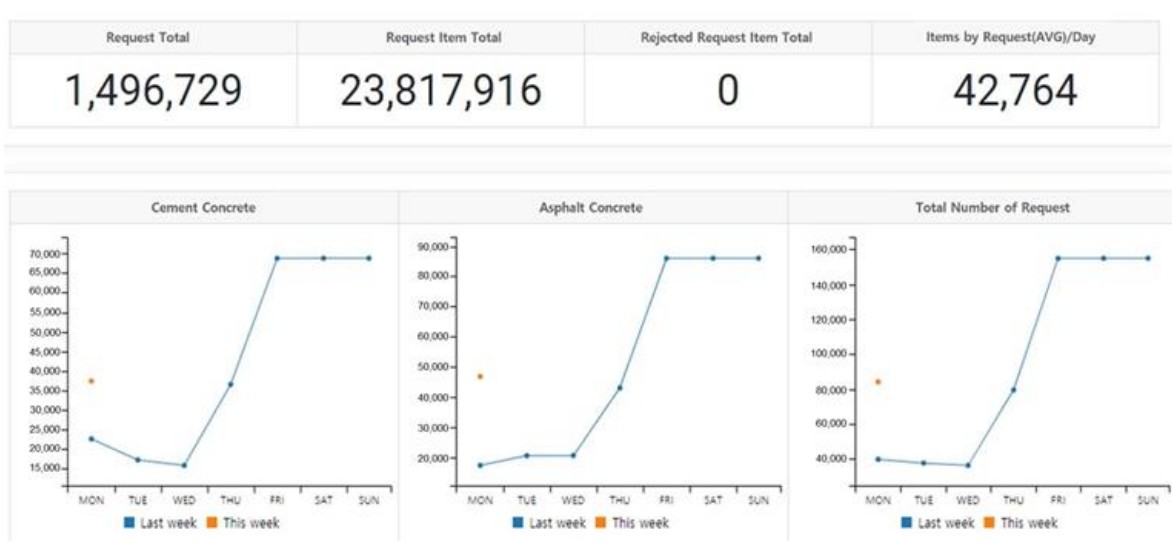

**Figure 7.** Dashboard of proposed system.

### 3.4. Implementation of IoT Collection Server Virtualization Scale-Out

For the proposed system, virtualization based on cloud technology was employed to operate a distributed system and multiple servers. In addition, adaptive server scale-out was implemented to efficiently manage resources in the system architecture and operation. Server virtualization was applied and implemented for the IoT sensor data collection process to make the traffic for data collection and processing suitable for the variable road pavement quality management environment. Figure 8 shows that the virtualization server is scaled out according to traffic and returns to its original state as traffic decreases in the

implemented system. This can increase big data processing performance because it enables efficient data management and system resource management.

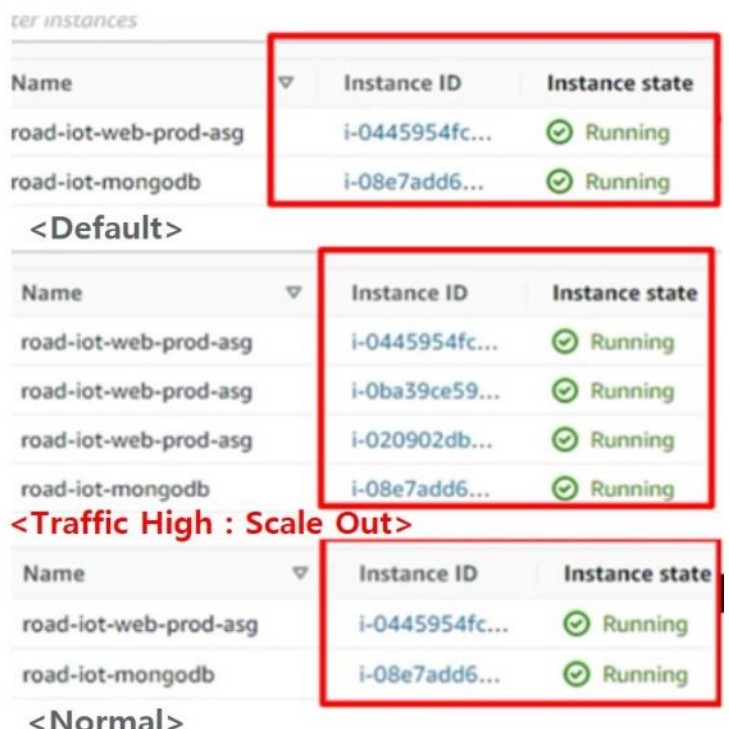

**Figure 8.** Virtualization server scale-out.

## 4. Experiment and Result

### 4.1. Experimental Setup

To verify the performance of the proposed system, the data collection and search performance were verified through experiments using MySQL, a general conventional RDBMS, and MongoDB, which is typically used in NoSQL. The data write performance was tested to verify the ability to collect large amounts of data in real time, and the read performance was tested to verify the search performance required for data analysis. The hardware environment and version of each software used in this experiment were as follows:

- CPU: Intel Core i9-8950HK 2.90GHz
- RAM: 32GB
- RDBMS—MySQL version: 8.0.22 [28]
- NoSQL—MongoDB version: 4.4.1 [29]

The structure of the dataset used in this experiment is as follows. Virtual data were created using assumptions regarding the data collected based on the location of each device, given that the data can occur in an actual road pavement environment. Then, through experiments using these data, the levels of performance of the write, read, and delete operations were verified.

- ID: Integer
- DeviceID: String
- Speed: Integer
- FlatnessL: Double
- FlatnessR: Double
- Longitude: Double
- Latitude: Double

A virtual dataset was constructed via random generation of values. Portions of the experimental data used are shown in Figures 9 and 10.

```
_id: ObjectId("5f9b020bace02d0e861b4620")
key: 0
deviceID: 68
longitude: 11.025278811623357
latitude: 127.28227578381937
speed: 74
flatnessL: 0.20285924509093922
flatnessR: 0.22590771752335947

_id: ObjectId("5f9b020bace02d0e861b4621")
key: 1
deviceID: 38
longitude: 30.85329159908171
latitude: 1.4599927938782964
speed: 58
flatnessL: 0.7503114643031159
flatnessR: 0.7598595695613977
```

**Figure 9.** NoSQL Data Sample.

| ID | DEVICE_ID | LONGITUDE | LATITUDE | SPEED | FLATNESSL | FLATNESSR |
|----|-----------|-----------|----------|-------|-----------|-----------|
| 0 | 47 | 83.11443184275512 | 36.39630969757356 | 6 | 0.635069 | 0.0798421 |
| 1 | 87 | 49.7441694329405 | 14.059090328723013 | 83 | 0.758301 | 0.561163 |
| 2 | 41 | 116.86195632018807 | 5.900622833055465 | 94 | 0.0808935 | 0.95301 |
| 3 | 71 | 15.280620705910204 | 41.13904217153135 | 38 | 0.0867338 | 0.661354 |
| 4 | 9 | 54.490355156947935 | 73.44568495958272 | 39 | 0.0485809 | 0.829282 |
| 5 | 48 | 0.24744617770996458 | 177.66091992809007 | 14 | 0.272452 | 0.166923 |
| 6 | 87 | 81.6989155806235 | 22.795559573000148 | 91 | 0.760368 | 0.246046 |
| 7 | 90 | 129.51522732944198 | 55.26562076911296 | 94 | 0.0743298 | 0.05887 |
| 8 | 39 | 11.608412094681523 | 163.292691489476 | 98 | 0.422637 | 0.990692 |
| 9 | 19 | 283.0335541077941 | 118.31208392274029 | 73 | 0.481868 | 0.89264 |
| 10 | 34 | 65.93684190610377 | 80.55084967849892 | 88 | 0.336781 | 0.973755 |
| 11 | 86 | 20.539425866964393 | 278.3147622438984 | 89 | 0.0921687 | 0.393268 |
| 12 | 10 | 104.352934277898 | 40.06275786141973 | 29 | 0.277596 | 0.75851 |
| 13 | 18 | 37.10818746639972 | 18.830702104556394 | 73 | 0.382597 | 0.00572497 |
| 14 | 69 | 52.730082689771535 | 44.44327120950698 | 88 | 0.698525 | 0.323135 |
| 15 | 72 | 182.79006681217498 | 51.66157236499736 | 24 | 0.915535 | 0.398916 |

**Figure 10.** RDBMS Data sample.

With regard to the total number of data points in the dataset used in this experiment, 100,000 data points were generated. The average time was calculated after each operation was performed five times. The unit of time is milliseconds (ms). In this experiment, the levels of performance of the write and read operations were tested using 10, 50, 100, 1000, 5000, 10,000, 20,000, and 50,000 data points. On the other hand, for the delete operation, the amounts of time required for deleting 10, 50, 100, 1000, 5000, and 10,000 data points corresponding to random keys were measured.

*4.2. Data Create (=Collect, Write) Operation Speed Experiment*

Table 1 shows the results of the write experiment.

For NoSQL, it took approximately 17 s to collect 50,000 data points, and 0.405 ms (0.000405 s) on average to collect 1 data point. By comparison, for MySQL, it took approximately 1027 s to collect 50,000 data points, and 21.146 ms (0.02146 s) on average to collect one data point. This result confirmed that the proposed system, the NoSQL-based collection system, improved performance by approximately 52 times compared to that of the conventional RDBMS-based system. Figure 11 shows a graph of the write experiment results for RDBMS and NoSQL DB. It can be observed in this graph that as the number of data points became larger, the performance difference increased rapidly.

**Table 1.** Results of write experiment.

| | NoSQL MongoDB | | RDBMS MySQL | |
|---|---|---|---|---|
| | Total Time (ms) | Unit Time (1 Time per ms) | Total Time (ms) | Unit Time (1 Time per ms) |
| 10 times | 9.4 | 0.9400 | 210.0 | 21 |
| 50 times | 16.8 | 0.3360 | 942.0 | 18.840 |
| 100 times | 34.6 | 0.3460 | 2135.6 | 21.356 |
| 1000 times | 351.4 | 0.3514 | 22,487.2 | 22.487 |
| 5000 times | 1576.4 | 0.3152 | 109,876.0 | 21.975 |
| 10,000 times | 3042.2 | 0.3042 | 214,680.8 | 21.468 |
| 20,000 times | 6152.6 | 0.3076 | 429,847.0 | 21.492 |
| 50,000 times | 17,074.8 | 0.3414 | 1,027,717.6 | 20.554 |

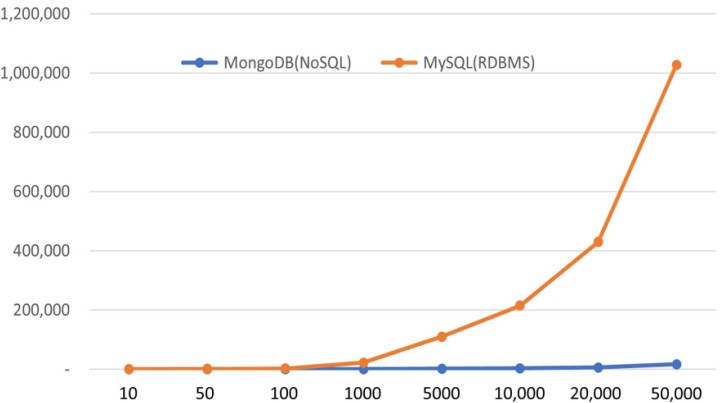

**Figure 11.** Results of write performance experiment (*x*-axis: number of data points, *y*-axis: write speed in ms).

*4.3. Data Read (=Search) Operation Speed Experiment*

Table 2 shows the results of the read operation experiment. In this experiment, a data reading operation was performed, in which specific values were searched for among existing stored data. The measurements obtained herein can be used as a performance index for analysis, statistics, and data searches. It took approximately 14.306 s for NoSQL to search and read 50,000 data points from 100,000 data points, and 0.299 ms (0.000299 s) on average to read 1 data point. By comparison, it took approximately 13.663 s for MySQL to search and read 50,000 data points from 100,000 data points, and 0.306 ms (0.000306 s) on average to read 1 data point. It can be observed that the search and read performance of the proposed system, the NoSQL-based system, was highly similar to that of the conventional RDBMS-based system.

**Table 2.** Results of read experiment.

| | NoSQL MongoDB | | RDBMS MySQL | |
|---|---|---|---|---|
| | Total Time (ms) | Unit Time (1 Time per ms) | Total Time (ms) | Unit Time (1 Time per ms) |
| 10 times | 3.2 | 0.320 | 3.8 | 0.380 |
| 50 times | 15.8 | 0.316 | 18.6 | 0.372 |
| 100 times | 29.8 | 0.298 | 32.4 | 0.324 |
| 1000 times | 298.0 | 0.298 | 286.8 | 0.286 |
| 5000 times | 1516.6 | 0.303 | 1332.8 | 0.266 |
| 10,000 times | 2887.0 | 0.288 | 2691.4 | 0.269 |
| 20,000 times | 5766.4 | 0.288 | 5661.0 | 0.283 |
| 50,000 times | 14,306.8 | 0.286 | 13,663.2 | 0.273 |

Figure 12 shows a graph of the read experiment results for RDBMS and NoSQL. This graph also shows that the two systems did not have a large difference in their performance.

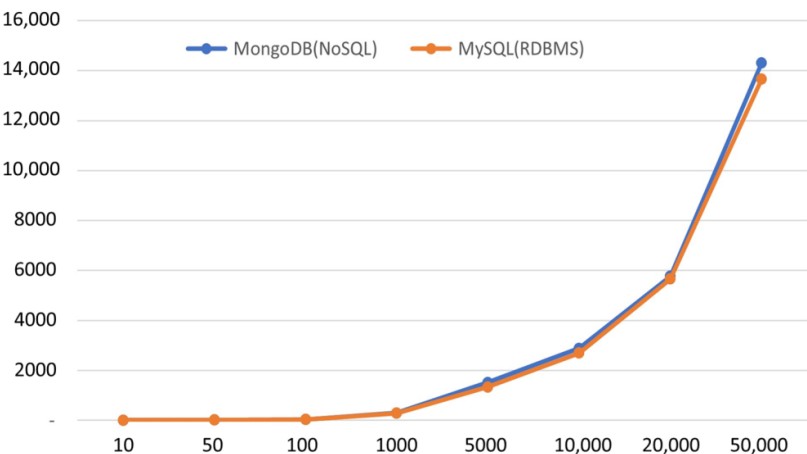

**Figure 12.** Results of read performance experiment (*x*-axis: number of data points, *y*-axis: read speed in ms).

*4.4. Data Delete Operation Speed Experiment*

Table 3 presents the results of the delete operation experiment. In this experiment, certain values were searched for among the existing stored data, after which the searched-for data were deleted. It took approximately 175.139 ms for NoSQL to search and delete 10,000 data points, and 15.909 ms (0.015909 s) on average to delete 1 data point. By comparison, it took approximately 558.696 s for MySQL to search and delete 10,000 data points, and 74.763 ms (0.07476 s) on average to delete one data point. This shows that the proposed system, the NoSQL-based search/delete system, improved performance by approximately 4.7 times compared to that of the RDBMS-based system. Figure 13 shows a graph of the delete experiment results for RDBMS and NoSQL. It can be observed that as the number of data points increased, the performance difference also increased rapidly.

**Table 3.** Results of delete experiment.

| | NoSQL MongoDB | | RDBMS MySQL | |
|---|---|---|---|---|
| | Total Time (ms) | Unit Time (1 Time per ms) | Total Time (ms) | Unit Time (1 Time per ms) |
| 10 times | 190.6 | 19.060 | 886.8 | 88.680 |
| 50 times | 945.2 | 18.904 | 5159.4 | 103.188 |
| 100 times | 1857.6 | 18.576 | 9800.0 | 98.000 |
| 1000 times | 18,352.4 | 18.352 | 95,250.2 | 95.250 |
| 5000 times | 93,293.8 | 18.658 | 410,245.4 | 82.049 |
| 10,000 times | 17,5139.8 | 17.513 | 558,696.8 | 55.869 |

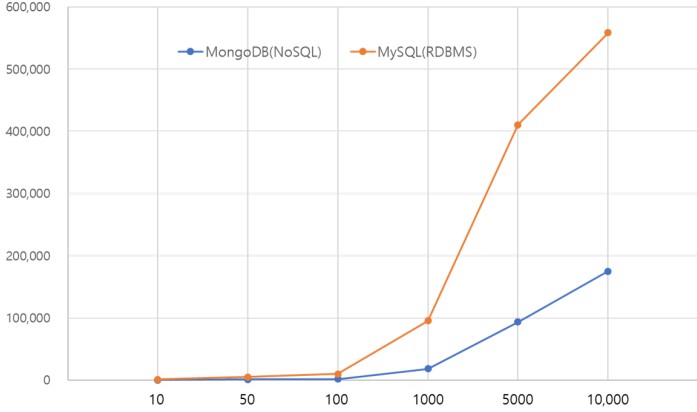

**Figure 13.** Results of delete performance experiment (*x*-axis: number of data points, *y*-axis: read speed in ms).

*4.5. Analysis of Experiment Results*

According to the experimental results, the write performance of MongoDB was 0.405 ms per data point on average, whereas that of MySQL was 21.146 ms per data point on average. Thus, MongoDB showed excellent performance with a significant difference from that of the comparison system. It can clearly be observed that for write operations, MongoDB was approximately 52 times faster than MySQL. On the other hand, the read performance of MongoDB was 0.299 ms per data point on average, whereas that of MySQL was 0.306 ms per data point on average. Because the two DBs exhibited only a slight difference in this regard, the levels of their search performance can be considered to be similar. However, MongoDB is flexible in data search because it is composed of a document format, and thus it is easier to improve and expand the performance because the database can easily interoperate with document search engines such as ElasticSearch. Meanwhile, the delete performance of MongoDB was 15.909 ms per data point on average, whereas that of MySQL was 74.763 ms per data point on average. Thus, MongoDB demonstrated excellent performance with a large difference in speed and, in delete operations, was approximately 4.7 times faster than MySQL.

The overall performance test results showed that the create, read, update, and delete (CRUD) performance of MongoDB, which is a NoSQL DB, is higher than that of MySQL, which is an RDBMS. This performance difference can be considered to be due to the differences in data structure and query structure of NoSQL and RDBMS, even though the DB models themselves may have a performance difference.

In this study, each DB was tested for very simple data query performance, and techniques such as indexing and distributed processing were not applied. However, it can be observed that the NoSQL DB demonstrated a better general performance. Therefore, it can be determined that a NoSQL DB will be more suitable for the big data platform for data collection, processing, and analysis.

*4.6. Analysis of IoT Sensor Traffic of Proposed System in Road Pavement Quality Management Environment*

To collect information on actual road pavement quality factors, we constructed the proposed system based on NoSQL, including server virtualization. The DB was implemented based on MongoDB using Kubernetes [30]. IoT sensors were installed on an actual road pavement site, after which data collection was performed using the implemented API. These data were stored in the NoSQL DB, and the system was implemented such that the virtualization server scale-out would be performed according to the traffic. Figures 14 and 15 show the traffic of the data collected through the actual system. Approximately 80,000 data points were collected per day, and approximately 50 to 60 data points were generated per minute. Figure 16 shows the traffic generated for each road pavement factor. These results confirm that the proposed system can efficiently collect IoT sensor data that are frequently generated and can perform big data processing using its NoSQL-based high-speed processing and server-scale-out technologies, even if the traffic increases.

In conclusion, it was confirmed that the NoSQL-based IoT data analysis system had high scalability and fast processing speed. Through comparison experiments with an existing RDBMS, it was possible to confirm the faster processing speed and flexible scalability of the proposed system. For an environment where large amounts of IoT data flow, the proposed system structure is judged to be a system suitable for high-speed processing and data warehouse expansion.

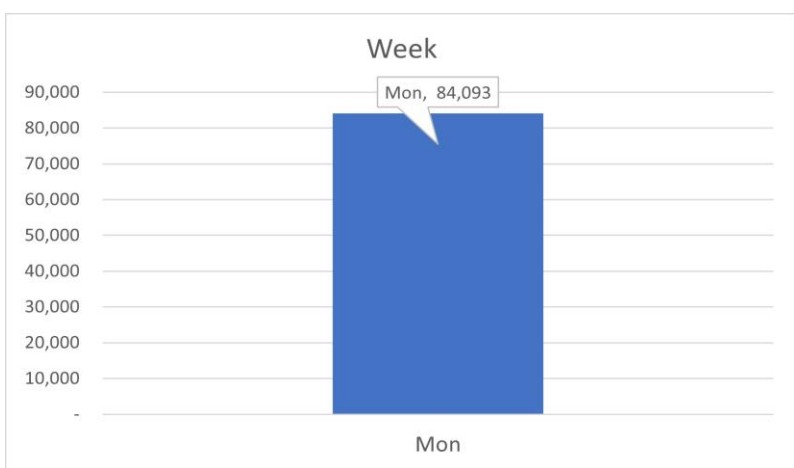

**Figure 14.** Traffic per day.

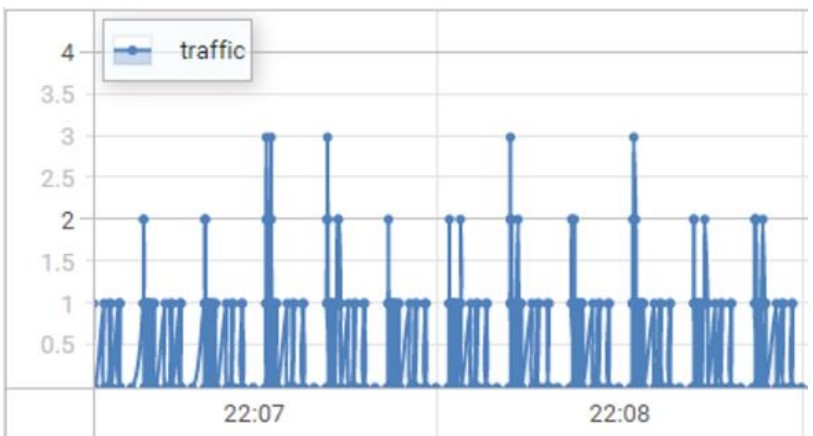

**Figure 15.** Traffic per minute.

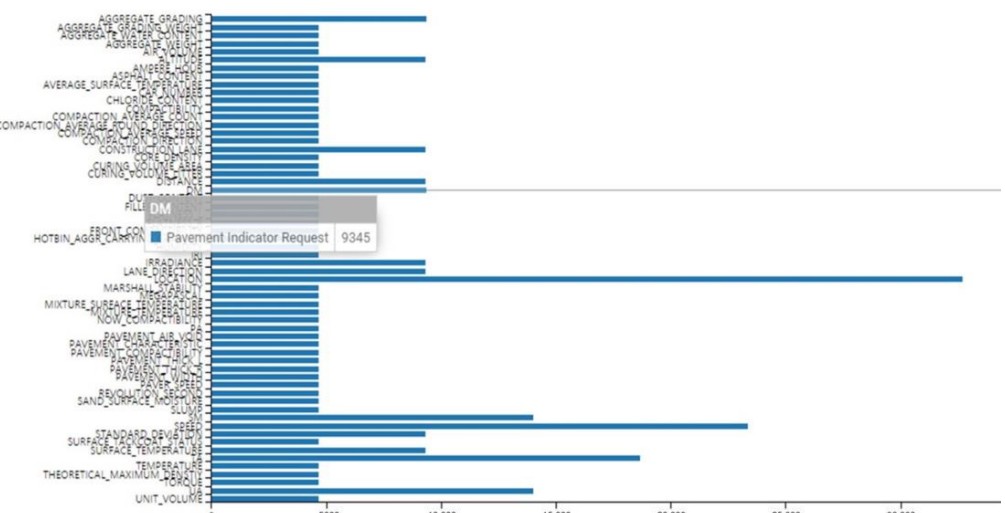

**Figure 16.** Traffic with respect to pavement quality parameters.

## 5. Conclusions

This study proposed components for a pavement quality management platform for the implementation of quality management of work processes in road pavement construction. Although an RDBMS structure for general alphanumeric data processing may be used for these purposes, it is limited by disadvantages in its performance. Instead, given

that real-time large-capacity data in various data formats are collected using a number of heterogeneous sensors that gather and process a variety of road pavement quality information, a structure based on NoSQL (MongoDB) was proposed. Additionally, a comparative experiment was conducted using data from an actual road pavement construction site.

The proposed system for road pavement quality management and analysis is a road pavement big data system for storing, processing, and analyzing location-based data and sensing data collected from IoT sensors. The proposed system can perform monitoring in real time for quality and process management, which thus far has been managed offline. The proposed system performs better than RDBMS-based IoT management platforms because it is constructed with cloud and NoSQL-based data warehouses. This research can contribute academically by providing a method for securing the real-time properties of IoT sensors, which, to the best of our knowledge, has not been performed in previous studies.

The proposed system can flexibly collect, process, and manage heterogeneous data generated in many construction sites, including road pavement sites where big data are generated, and even perform analysis and monitoring using these data. It was verified through experiments that the proposed system had a higher performance than those of existing systems.

Furthermore, the proposed system is suitable for application to road pavement quality management platforms and is easy to expand because it uses MongoDB, which has high interoperability with the open-source search engine ElasticSearch and the distributed processing platform Hadoop. Interoperation with ElasticSearch and Hadoop is expected to enable the construction of a big data platform that can perform real-time data collection and monitoring and high-speed data processing by improving distributed and parallel processing performance. Therefore, the big data analysis system that is currently designed with ElasticSearch/Hadoop/MongoDB is believed to be highly suitable for data collection, processing, and analysis. Moreover, the system was designed to efficiently operate the server and resources according to variable traffic using server virtualization scale-out. This is expected to be expandable as a business model as construction site management platforms are established in the future.

To verify the proposed system, we implemented the proposed system and collected data using IoT sensors from a real road pavement site in Korea. The interface API was implemented using the proposed API structure, a NoSQL-based database was constructed, and then data were collected in real time. The implementation results demonstrate that the proposed system can collect data in real time and can perform monitoring, analysis, and visualization. The proposed IoT location-based smart platforms for pavement quality by using a NoSQL based system (create: 0.405 ms, read: 0.299 ms, delete: 15.909 ms) was significantly higher than that of the compared RDBMS based system (create: 21.146 ms, read: 0.306 ms, delete: 74.763 ms). Real-time processing capacity was also verified based on quality big data generated on actual road pavements, and the proposed system was proven suitable for the real-time monitoring (the data collection processing) of road pavement quality big data.

In the road pavement process, the road pavement quality big data can be measured according to the state of the road pavement material and the progress of each unit process by using the large amount of sensor data during the very long construction period. Therefore, a system capable of real-time monitoring and big data processing is necessary for managing the road pavement quality big data. The proposed framework can respond immediately to deal with the big data related materials or processes, and can check the status of the road pavement quality in real time. A NoSQL-based data platform was built for real-time big data processing. This is a structure that can be easily expanded even if additional data collection is required. In particular, in the road pavement process affected by external environmental factors, a big data system capable of real-time monitoring can be used as a key technology for process quality control.

In the future, we plan to research technologies that can improve the efficiency of process management by analyzing the road pavement quality level through machine

learning based on collected data, determining the quality level, and predicting changes in the quality level.

**Author Contributions:** Conceptualization, S.-S.H., J.L. and B.K.; data curation, S.-S.H.; formal analysis, S.-S.H.; investigation, J.L.; methodology, S.-S.H.; project administration, B.K. and J.L.; resources, S.C.; software, S.-S.H.; supervision, B.K. and J.L.; validation, S.-S.H.; visualization, S.C.; writing—original draft, S.-S.H.; writing—review & editing, S.C. All authors have read and agreed to the published version of the manuscript.

**Funding:** This work is supported by the Korea Agency for Infrastructure Technology Advancement (KAIA) grant funded by the Ministry of Land, Infrastructure, and Transport (Grant 22POQW-C152690-04).

**Institutional Review Board Statement:** Not applicable.

**Informed Consent Statement:** Not applicable.

**Data Availability Statement:** Not applicable.

**Conflicts of Interest:** The authors declare no conflict of interest.

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
