# Peer review of "Fast Real-Time Data Process Analysis Based on NoSQL for IoT Pavement Quality Management Platform"

_applsci, doi:10.3390/app13010658_

Round 1
Reviewer 1 Report
This study presents a fast real-time data processing analysis framework by employing the NoSQL database and IoT-based pavement quality management platform. The database management system (RDBMS)-based infrastructure is designed to enable real-time high-speed collection and analysis by applying a NoSQL-based database, and a big data platform was designed for data collection and visualization infrastructure. The paper is well organized, and can be published after following issues are reoslved.
(1) The theoretical contributions are missing, and it will be better if theoretical novelty is addressed.
(2) The comparison analysis is necessary to vadilate the advantages of proposed framework.
(3)The practical applications need to be further discussed to verifty this process.
(4) Discussions on the theoretical contributions and managerial insights should be added.
Reviewer 2 Report
This paper proposed an IoT system based on NoSQL for real-time application. The results show that the performance has improved significantly in deleting data.
Author Response
Dear Reviewer,
Thank you for reviewing our research and for your kind comments.
Thank you for taking your time.
Sincerly,
Sung-Sam Hong
Jangan University.
Reviewer 3 Report
This work illustrates a data collection and analysis system based on sql and nosql databases and a distributed and parallel analysis. The system monitors and analyzes real-time data relating to the quality of road surfaces.
In section 2.1 it would be better to deepen the typology of sensors applied in the various studies in terms of measured variable and sampling frequencies, highlighting the pros and cons of the selected sensors
Why is evidence of the characteristics of NoSQL databases (Section 2.2) present in the "Related Works" section? It would be preferable to include it in the description of the infrastructure implemented or in a separate section dedicated to ICT technologies
Section 3 is well structured and clarifies the structure and functioning of the system well; the label relating to figure 5 should be repositioned and integrate the section 2.2 if not added in a new section.
Section 4 details the comparisons made in CRUD operations between the NoSQL database and SQL. It may be important to explain the exclusion of time-serious databases from this comparison, especially databases such as InfluxDB or Apache IoTDB.
The first time an acronym is used it is better to specify it in full (line 47 PMS)
Round 2
Reviewer 1 Report
The paper is well organized, and it can be published after following issues are resolved:
1. It lacks motivations and research gap from the theoretical veiwpoint.
2.It is not enough for the current literature review part.
3.There is little comparison analysis on results, and how to verify the advantages of the proposed f IoT location-based smart platforms.
